# Long-Term Suppression of c-Jun and nNOS Preserves Ultrastructural Features of Lower Motor Neurons and Forelimb Function after Brachial Plexus Roots Avulsion

**DOI:** 10.3390/cells10071614

**Published:** 2021-06-28

**Authors:** Prince Last Mudenda Zilundu, Xiaoying Xu, Zaara Liaquat, Yaqiong Wang, Ke Zhong, Rao Fu, Lihua Zhou

**Affiliations:** 1Department of Anatomy, Zhongshan School of Medicine, Sun Yat-sen University, Guangzhou 510080, China; pzilundu@gmail.com (P.L.M.Z.); xuxiaoying_2001@163.com (X.X.); zhongk5@mail2.sysu.edu.cn (K.Z.); 2Department of Anatomy, School of Medicine, Sun Yat-sen University, Shenzhen 518100, China; liaquat@mail2.sysu.edu.cn; 3Department of Electron Microscopy, Zhongshan School of Medicine, Sun Yat-sen University, Guangzhou 510080, China; wangyaq2@mail.sysu.edu.cn

**Keywords:** motor neuron ultrastructure, ventral horn, brachial plexus roots avulsion, neuroprotection, motor function

## Abstract

Brachial plexus root avulsions cause debilitating upper limb paralysis. Short-term neuroprotective treatments have reported preservation of motor neurons and function in model animals while reports of long-term benefits of such treatments are scarce, especially the morphological sequelae. This morphological study investigated the long-term suppression of c-Jun- and neuronal nitric oxide synthase (nNOS) (neuroprotective treatments for one month) on the motor neuron survival, ultrastructural features of lower motor neurons, and forelimb function at six months after brachial plexus roots avulsion. Neuroprotective treatments reduced oxidative stress and preserved ventral horn motor neurons at the end of the 28-day treatment period relative to vehicle treated ones. Motor neuron sparing was associated with suppression of c-Jun, nNOS, and pro-apoptotic proteins Bim and caspases at this time point. Following 6 months of survival, neutral red staining revealed a significant loss of most of the motor neurons and ventral horn atrophy in the avulsed C6, 7, and 8 cervical segments among the vehicle-treated rats (*n* = 4). However, rats that received neuroprotective treatments c-Jun JNK inhibitor, SP600125 (*n* = 4) and a selective inhibitor of nNOS, 7-nitroindazole (*n* = 4), retained over half of their motor neurons in the ipsilateral avulsed side compared. Myelinated axons in the avulsed ventral horns of vehicle-treated rats were smaller but numerous compared to the intact contralateral ventral horns or neuroprotective-treated groups. In the neuroprotective treatment groups, there was the preservation of myelin thickness around large-caliber axons. Ultrastructural evaluation also confirmed the preservation of organelles including mitochondria and synapses in the two groups that received neuroprotective treatments compared with vehicle controls. Also, forelimb functional evaluation demonstrated that neuroprotective treatments improved functional abilities in the rats. In conclusion, neuroprotective treatments aimed at suppressing degenerative c-Jun and nNOS attenuated apoptosis, provided long-term preservation of motor neurons, their organelles, ventral horn size, and forelimb function.

## 1. Introduction

Brachial plexus root avulsion is a leading cause of debilitating chronic pain and upper limb paralysis [1]. It involves spinal nerve root traction and avulsion followed by inflammation, invasion by glial cells, and motor neuron apoptosis which occur in the spinal cord [2]. Previous studies have reported Nissl substance dispersion, organelle dysmorphia, synaptic stripping, glial cell over-activation, downregulation of acetylcholinesterase immunoreactivity as well as a profound functional loss within the first month post avulsion and up to three months [2,3,4,5,6].

However, these studies have described the morphological, qualitative, and functional changes recorded over shorter periods not exceeding 90 days [7,8]. In reality, though, some patients get brachial plexus injury treatment long after the primary injury [9]. Furthermore, some treatments initiated late after avulsion still showed some functional changes at the organismic level [9,10,11,12,13,14,15]. A knowledge gap, therefore, exists in the literature on the mechanism and cellular ultrastructure after a longer time following brachial plexus roots avulsions, especially after some neuroprotective treatments. Long-term ultrastructural features of the surviving spinal cord cells must be understood since these cells and their surroundings can serve as targets of delayed treatments in human patients and model animals. Treatments such as neurotrophic delivery, cell transplantation, use of scaffolding materials, nerve grafting, re-implantation, and nerve transfers do not fully prevent long-term upper limb paralysis [16,17]. Therapies for brachial plexus root avulsions have been associated with poor clinical and functional outcomes owing to the low intrinsic regenerative capacity of the spinal cord, post-injury inhibitory milieu, and the fact that the majority of motor neurons undergo apoptosis within the first month after avulsion [6].

The protein products of the JUN and NOS1 genes have been implicated in the occurrence of motor neuron apoptosis post-brachial plexus injury. Previously, double immunostaining showed induction and colocalization of the immediate early gene transcription factor subunit c-Jun and the nitric oxide generating neuronal nitric oxide synthase (nNOS) proteins within the large presumptive alpha motor neurons post avulsion injury [6]. C-Jun was immediately overexpressed, peaks at 3 days and sustained above baseline levels while nNOS protein expression does not appear until the end of the first week peaking at the end of the second week and then falls at 28 days [6]. C-Jun, a major subunit of the activating protein 1 (AP-1) transcription factor, and its phosphorylation product p-c-Jun have been implicated in cellular apoptosis [18,19,20]. Pharmacological inhibition of overexpressed c-Jun using neuroprotective treatments such as reversible ATP-competitive inhibitor JNK inhibitor SP600125 was neuroprotective to motor neurons of the spinal cord and retinal ganglion cells after avulsion [19,21,22].

nNOS, one of the three isoforms that generate nitric oxide from molecular oxygen and L-arginine, has also been implicated in nitric oxide-related oxidative stress that is neurodegenerative when in excess [23]. Pharmacological suppression of nNOS using 7-nitroindazole was shown to be neuroprotective in various spinal cord injury models including brachial plexus avulsion [21,24,25,26,27]. However, these neuroprotective observations were all restricted to shorter time points thereby creating a knowledge gap on the long-term relevance of such preclinical interventions. In this study, we used pharmacological methods to inhibit c-Jun (JNK inhibitor SP600125) and nNOS (7-nitroindazole) over a month during which most of the motor neurons are known to die and compared the ultrastructural features of the injured motor neuron cytosolic, mitochondrial, and nuclear compartments 6 months post-injury. This would address a dearth of literature on the long-term state of the ventral horn area and cellular ultrastructure therein following brachial plexus roots avulsion, with or without treatment, beyond three months in rats. The study also sought to evaluate the effects of chronic pharmacological inhibition of c-Jun and nNOS on oxidative/nitrosative stress and markers of the candidate mitochondrial motor neuron apoptosis-signaling pathway.

Transmission electron microscopy provides a high-resolution snippet into the ultrastructural features of cells and allows inspection of physical changes associated with avulsion injuries and therapies. It also helps identify pertinent sites of the ultrastructural changes that are associated with prominent signaling pathways that could underpin neurodegeneration or neuroprotection. Therefore, the current study used electron microscopy to evaluate the ultrastructural changes in the ventral horn cells at six months after brachial plexus roots avulsion. Thus, the data herein provide morphological evidence and help illuminate brachial plexus roots avulsion-induced chronic spinal cord injury.

## 2. Materials and Methods

### 2.1. Experimental Animals and Ethics Statement

Forty-two adult Sprague Dawley rats (180–250 g) were purchased from the Laboratory and Experimental Animal Center of Sun Yat-sen University. The rats were housed under a 12-h light/dark cycle, with unlimited access to rat chow and water. The Sun Yat-sen University Animal Experimentation Ethics Committee approved all experimental procedures which were in accordance with the Chinese Guidelines for the Humane Treatment of Laboratory Animals. The rats’ well-being was observed throughout the experimental period and all efforts to minimize the use of animals were made.

#### Brachial Plexus Avulsion

The brachial plexus avulsion model was performed on the right side of twelve rats according to an established protocol [6]. The surgical operations were performed aseptically under Ketamine (50 mg/mL) and Xylazine (20 mg/mL) general anesthesia which was given intraperitoneally. The rats were secured in a prone position under a dissection microscope; a straight incision, about 4 cm long was made along the midline of the neck from the occiput to the level of the upper corner of the scapula. The muscles of the neck were pulled outward and the small muscles adhering to the spinous processes as well as vertebral lamina were stripped off to expose the articular processes of C4-T1. Laminectomy on the right side of C6-T1 was performed using a vertebral lamina rongeur. The dura- and subarachnoid mater were dissected and opened to expose the dorsal and ventral roots of the C6, C7, and C8 spinal nerves. After identification, all the dorsal and ventral rootlets of C6, C7, and 8 were pulled out using a custom-made micropipette hook. Muscles, fascia, and skin were then sutured successively in layers and the rats were allowed to survive for 6 months. They were cared for until their scheduled sacrifice times of 28 days and 180 days.

### 2.2. Experimental Grouping and Treatments

The brachial plexus roots avulsed rats were randomly divided into a vehicle control group (coconut oil), JNK inhibitor SP600125 (25 mg/kg dissolved in coconut oil) group, and a neuronal NOS inhibitor 7-nitroindazole (30 mg/kg dissolved coconut oil) group. There were four rats in each of the electron microscopy groups to minimize animal use. For the 28-day chronic inhibition study, each group had 10 rats. Avulsion causes apoptotic death of the majority of ventral horn motor neurons within the first month [6]. Therefore, treatments were given over 28 days by daily intraperitoneal injections. At this point, 10 rats from the vehicle, SP600125 and 7-nitroindazole-treated groups were sacrificed for Western blotting (*n* = 5) and immunofluorescence imaging (*n* = 5) evaluations. The rats underwent functional evaluation before and every week over 28 days, and then at 3 and 6 months post-avulsion. Upon reaching 28 days, 10 rats from each group were terminally injected with an intraperitoneal overdose of Ketamine (50 mg/mL) and Xylazine (20 mg/mL) and then immunofluorescence imaging perfused transcardially with normal saline followed by precooled 4% paraformaldehyde. The spinal cord segments were harvested and post-fixed in 4% paraformaldehyde for 3–4 h and then in 30% sucrose for another 2–4 days or until they sank to the bottom of the EP tube. Fresh tissue harvesting was performed for the Western blotting assay. For the long-term electron microscopy study (after 6 months post avulsion), all four rats from each group were terminally injected with an intraperitoneal overdose of Ketamine (50 mg/mL) and Xylazine (20 mg/mL) and then perfused transcardially with 2.5% glutaraldehyde paraformaldehyde-glutaraldehyde fixative in phosphate buffer (pH 7.4). The rats were once again dissected and C6–8 spinal cord segments removed for morphological studies.

### 2.3. Electron Microscopy

The cervical spinal cord segments were post-fixed by immersion in a mixture of 2.5% paraformaldehyde–glutaraldehyde fixative for 24–72 h. After sectioning, 35 μm slices of C7 segments from each group were selected and washed thrice using 0.1 mol/L phosphate buffer and 1% osmium tetroxide in 0.1 mol/L Na-phosphate (pH 7.3) buffers. The selected segments were dehydrated in a graded concentration of ethanol followed by propylene oxide and embedded in epon plastic resin. The Ultrotom VRV (LKB Bromma, Berlin, Germany) was used to cut 70 nm ultrathin sections which were then stained with uranyl acetate and lead citrate and examined on a Tecnai™ Spirit transmission electron microscope (Fei, Osaka, Japan) at Sun Yat-sen University’s Electron Microscopy Unit.

### 2.4. Analysis of the Ultrathin Sections

Large neurons, approximately 35 μm wide, with a nuclear plane showing and C-type nerve terminals visible were designated alpha motor neurons [28]. The cell surfaces were inspected for completeness and then its perimeter captured at a magnification of 18,500× using a digital camera attached to the Tecnai™ Spirit transmission electron microscope (Fei, Japan). The perimeter of the complete neurons was measured and the number of synaptic terminals apposing the motor neuron somata was recorded per every 100 μm of the cell membrane by assistants blinded to the treatment conditions. Three types of synaptic input terminal were recognized under high magnification (at least 25,000×) according to the [29] nomenclature: F-type (flattened synaptic vesicles and are inhibitory), S-type (spherical synaptic vesicles and are excitatory inputs), or C (cholinergic inputs). In total, five neurons per rat in each group were examined (*n* = 20 neurons per group: 7-nitroindazole, SP600125, and vehicle control rat. The visual contrast between some structures on the same electron micrograph was digitally colored with the Adobe Photoshop software to make them more visible.

### 2.5. Morphometrical and Statistical Analysis

The morphometric studies were performed on the thin sections and transmission electron microscope images of cervical segments of both control and treated rats. Research assistants were blinded to the treatment condition of the rats. Mitochondrial density and size were assessed on four random 16 μm^2^ fields on 3900× TEM (transmission electron microcope) images of each motor neuron using Image J software (1.53j, National Institutes of Health, Bethesda, MD, USA). The pixel measurements were taken of length (longest point) and width (widest point) of the mitochondria in view using Image J software. The measurements were carried out by two assistants blinded to the treatment conditions and their results did not differ significantly. As described previously [30], an additional magnification correction factor was not used for output from either microscope because the difference between an imaged micrometer with microscope-stated versus microscope-calculated magnification was negligible (~2%).

Mitochondrial cristae appearance was graded by using a five-point scale, 1 for the most degenerated or absent-appearing cristae and 5 for the most intact and well ordered. Observers were trained to use the scale and checked each other over one negative per animal to ensure conformity [30].

The extent of the axonal pathology near the motor neurons was evaluated using a semi-quantitative scoring system adapted from [3,31]. Ali and colleagues (2016) noted that direct counting of axons is very difficult and standard approaches using percentage area and profile counting fail to provide meaningful quantification of individual axon numbers hence this method [3]. Specifically, examinations were performed on spinal cord electron micrograph sections at magnification 3900× showing an alpha motor neuron (5 neurons per rat) of the C7 by assistants blinded to the treatment conditions of the groups. Semi-quantitative analysis scores, adapted from [3,31] were rated as follows. 0 = absent; 1 = minimal (fewer numbers of small axons, myelin sheath disruption and debris; 2 = moderate (moderate mix of the numbers of small myelinated and non-myelinated axons, myelin sheath disruption, axon shrinkage, edema, and debris); or 3 = extensive (very high numbers of small axons, unmyelinated, myelin disruption, shrinkage, edema, and debris as well as collagen deposits in endoneurium).

### 2.6. Light Microscopy: Motor Neuron Survival

Neutral red staining (1%), following a previous protocol [21] was used to evaluate the motor neuron survival 6 months after avulsion. Serial axial spinal cord sections from C6, C7, and C8 segments were selected (350 μm apart; approximately 10 sections per spinal cord level) and all neutral red stained motor neurons with visible nuclei were counted on both the intact and lesioned sides of the C6, C7, and C8 spinal segments. The number of motor neurons on the contralateral side was expressed as 100% so that the number of surviving motor neurons on the ipsilateral side was expressed as its fraction. We also evaluated the ventral horn size (area by planimetry) on each of the above selected neutral red stained C6–8 cervical cord segment slices 6 months after avulsion using the MountainsLab™ area calculation tool. The size of the ipsilateral ventral horn was normalized to the corresponding contralateral side at the same level.

### 2.7. Immunofluorescence

The immunofluorescence procedures were performed according to our previous publication [32]. Briefly, the C6, C7, and C8 spinal segments of each rat were removed carefully, immersed in fixative (4% paraformaldehyde solution) for about 3–4 h, and then suspended in 30% (*v*/*v*) sucrose solution in phosphate-buffered saline (PBS) until they sank to the bottom of the tube. Frozen transverse sections (35μm) were washed three times with 0.01M PBS for 10 min each time and incubated in 3% BSA 0.01M PBS at room temperature for 1 h. The slices were then incubated overnight at 4 °C with the following primary antibodies: Santa Cruz #3270, Santa Cruz Biotechnology Inc, Santa Cruz, CA, USA) c-Jun (1:200, Santa Cruz sc-74543), nNOS (1:200, Santa Cruz #4231), Glial fibrillary acidic protein (GFAP) (Santa Cruz Biotechnology Inc, Santa Cruz, CA, USA #sc-51908). After washing in PBS, sections were then incubated with respective secondary antibodies at room temperature for 1 h. The slices were then washed thrice, mounted on glass slides, and their images were captured using a Nikon Olympus BX-63 microscope. The staining specificity was tested by the omission of primary antibodies.

### 2.8. Western Blotting

The freshly obtained tissue or stored −80 °C from rats sacrificed at their respective time points were treated for Western blotting assay following previous publications [6]. Samples from five rats per group at the 28-day time point were pooled for further analysis of proteins of interest expression. Lysis buffer with a 1% cocktail of protease inhibitors and 1% phenylmethylsulfonyl fluoride (PMSF) (Nanjing KeyGen Biotech Co., Ltd., Nanjing, China) was added to the samples and all the steps were carried out on ice according to the kit manufacturer’s protocol. The spinal cord tissue samples were cut into small pieces using a heat sterilized and cooled pair of scissors and then sonicated on ice to a liquid. Protein concentration in each of the samples from different groups was determined using the Pierce™ BCA Protein Assay Kit (Thermo Fisher Scientific Inc, Waltham, MA, USA), according to the manufacturer’s protocol. All the samples per group were diluted using an equal volume of the 5X SDS loading buffer. 40 µg of proteins from each sample were loaded and separated using a 10% TGX™ FastCast™ Acrylamide kit (Bio-Rad Laboratories Inc., Hercules, CA, USA) at 80 V for 1 h. The PageRuler™ Prestained Protein Ladder, 10 to 180 kDa (Thermo Fisher Scientific, USA) was used alongside the protein samples as the protein molecular weight standard. The resulting gel was then run on 80 V and 220 mA to electro-transfer the negatively charged proteins onto polyvinylidene difluoride (PVDF) membranes. The PVDF membranes were then blocked with 5% skim milk in Tris-buffered saline with 0.1% Tween^®^ 20 detergent (TBST) solution for 1 h at room temperature on a rocking platform. Next, the blocked PVDF membranes were probed with the following primary antibodies c-Jun (1:1000; Santa Cruz Biotechnology, Inc, Santa Cruz, CA, USA), nNOS (1:1000; Santa Cruz Biotechnology, Inc, Santa Cruz, CA, USA), Bim (1:1000; Santa Cruz Biotechnology, Inc, Santa Cruz, CA, USA), Cleaved caspase 3 (1:1000; Santa Cruz Biotechnology, Inc, Santa Cruz, CA, USA) and control GAPDH (1:5000; Sigma-Aldrich, St. Louis, MO, USA). The primary antibodies were diluted in Western Blot Immune Booster solution 1 (Santa Cruz Biotechnology, Inc., Santa Cruz, CA, USA) and incubated overnight at 4 °C. After 12 h, these antibody-bound membranes were washed thrice with TBST and then probed with respective horseradish peroxidase-conjugated goat anti-mouse IgG (1:5000; Merck KGaA, Darmstadt, Germany) or anti-rabbit IgG (1:5000; Merck KGaA, Darmstadt, Germany) secondary antibodies for 1 h at room temperature. The blots were washed thrice with TBST and then developed using the Pierce™ ECL Western Blotting Substrate (Thermo Fisher Scientific, Waltham, MA, USA) and exposed on the ChemiDoc XRS+ system (Bio-Rad Inc, Hercules, CA, USA). Exposed images were scanned, and protein bands quantified using the Image-J software (NIH, Bethesda, MD, USA).

### 2.9. Oxidative Stress: Nitric Oxide Production Assay

Spinal cord lysates from respective groups were used. The Total Nitric Oxide Assay Kit (Beyotime Biotechnology, China), was used to estimate nitric oxide production using the concentration of nitrate and nitrite in a modified Griess reaction assay according to the manufacturer’s protocol. Briefly, the respective sample solutions (60 μL/well) were added in a 96-well plate, followed by the sequential steps as outlined in the manual: addition of Nicotinamide adenine dinucleotide phosphate (NADPH) (2 nM), Flavin adenine dinucleotide (FAD), nitrate reductase, (incubation of mixture for 30 min at 37 °C), the addition of Lactate Dehydrogenase (LDH) Buffer, LDH, (incubation of mixture for 30 min at 37 °C). Lastly, the addition of Griess reagents I and II followed by mixing and a 10-min incubation at room temperature. The absorbance was then measured using the microplate reader at a wavelength of 540 nm and converted to micromoles per liter (μmol/L) using a standard curve that was generated by the addition of 0 to 80 μmol/L sodium nitrite (NaNO_2_) to fresh spinal cord tissue lysates. The injured ipsilateral side results were expressed as a percentage of the contralateral side.

### 2.10. Functional Evaluations

#### 2.10.1. Terzis Grooming Test

The Terzis grooming test was used to assess the gross motor function of the forelimbs according to a description by [33]. Briefly, each rat was acclimatized in a quiet environment and allowed to do trail runs over three days leading to final testing. The tests were performed at 28, 90 and 180 days post-avulsion. We gently sprayed 10% sucrose solution on the head, face, and neck of animals using a syringe to stimulate bilateral grooming behaviors involving the forelimbs. The right forelimb was scored based on the following criteria: 0—no response; 1—elbow flexion but unable to reach/touch nose; 2—elbow flexion and able to reach/touch nose; 3—elbow flexion and able to reach/touch the site below the eyes; 4—elbow flexion and able to reach the eyes; and 5—elbow flexion and able to reach/touch the ears or back of the ears [34].

#### 2.10.2. Grid Walk Test

The grid walk test was used to assess fine motor movements such as balance and forelimb placement at 3 and 6 months after brachial plexus avulsion following the methods described by [35]. Briefly, a one-meter-long runway with regularly assigned 3.5 cm gaps covering 80 cm was used. The rats were required to navigate the runway by carefully placing their forelimbs on the bars three times. Baseline training was repeated thrice over three days before the main test. The following scoring system was used to evaluate the rats’ ability to navigate the beams: number of errors out of the 20 grid and average of three trials for each rat (a maximum of 20 errors each trial). The numbers of errors were computed into the following scores: 0–1 error was rated as 3 points, 2–5 as 2 points, 6–9 as 1 point, and 10–20 footfalls as 0 points according to a previous study [36].

### 2.11. Statistical Analysis

All data were expressed as means ± standard deviation (SD) and statistically analyzed with SPSS v24 (IBM, Armonk, NY, USA) to compare different measurements of both control and treated animals. Multiple group comparisons were performed using one-way analysis of variance (ANOVA) and Tukey post-hoc test. The Kruskal Wallis H-test was used for ordinal data analysis from the grooming and grid walk tests. The differences were statistically significant if *p* < 0.05 and are depicted by asterisks *.

## 3. Results

### 3.1. Chronic Pharmacological Inhibition of c-Jun and nNOS Is Neuroprotective, Relieves Oxidative/Nitrosative Stress, and Promotes Functional Recovery

Motor neuron counting on neutral red stained slides revealed that 28-day courses of SP600125 and 7-nitroindazole were both significantly neuroprotective compared to vehicle treatment (ANOVA/Tukey all *p* < 0.05) (Figure 1A). In addition, 7-nitroindazole treatment was marginally superior to SP600125 in terms of motor neuron counts, although the difference was not statistically significant (*p* > 0.05) (Figure 1A). Neuroprotective treatments, SP600125 and 7-nitroindazole suppressed the expression of their respective targets, c-Jun and nNOS proteins at this time point when compared with the vehicle treatment in SD rats (Figure 1B i and ii; ANOVA/Tukey all *p* < 0.05). The neuroprotective treatments suppressed apoptosis of motor neurons by inhibiting the overexpression of components of the mitochondrial pathway of apoptosis namely Bim, and cleaved caspase 3 compared to that observed in vehicle-treated control rats (Figure 1B i and ii; ANOVA/Tukey all *p* < 0.05). In addition, treatment with SP600125 and 7-nitroindazole significantly reduced oxidative stress as indirectly measured by the total nitric oxide production compared with the controls (Figure 1C; ANOVA/Tukey all *p* < 0.05).

There was only slight atrophy of the ventral horns of the ipsilateral sides relative to the contralateral side at day 28 in both neuroprotective agent-treated groups while vehicle-treated rats showed significant atrophy (Figure 1D i and ii).

Taken together, these results provide evidence that chronic pharmacological inhibition of c-Jun and nNOS protein expression attenuated oxidative stress, apoptosis and preserved motor neurons at the end of one month. These benefits of chronic suppression coincide with the period during which the majority of motor neurons disproportionately die after brachial plexus roots avulsion.

### 3.2. Avulsion Causes Long-Term Ventral Horn Atrophy and Is Partially Rescued by Neuroprotective Treatments

Previous studies had documented major loss of the ventral horn motor neurons within a month and further at three months after avulsion in rats [6,21,37]. We evaluated the ventral horn size (area by planimetry) on neutral red stained C6–8 cervical cord segment slices 6 months after avulsion using the MountainsLab™ area calculation tool. The results showed no significant differences in the area sizes of the contralateral ventral horns between the control and treatment groups. Therefore, all results were expressed in relation to the contralateral side at the same level. The notable finding on light microscopy evaluation was the significant atrophy of the ipsilateral ventral horn in all rats irrespective of their treatment regime at 6 months after brachial roots avulsion. Although the sizes of the ventral horns of the spinal cord are not universally equal in size at each level, the vehicle-treated rats showed the highest level of atrophy in which the size of the ventral horns at all segments under study [C6–8] and was reduced by nearly a third (ANOVA *p* < 0.05; Tukey’s post hoc *p* < 0.05). The 7-nitroindazole-treated group had the least atrophy followed by the SP600125 treatment group (Figure 2A,B), although the difference between these two neuroprotectant-treated groups were not statistically different (*p* > 0.05).

### 3.3. Avulsion Causes Motor Neuron Loss at 6 Months Post-Injury and Was Attenuated by Neuroprotective Treatments

Neutral red staining was carried out to assess residual motor neuron survival at 6 months after avulsion on 10 slices from each of the three segments (C6–8) in each rat following a previously described method [21]. There were no significant differences in contralateral sides motor neuron counts between groups, as a result, the contralateral side was used as a standard upon which the ipsilateral side motor neuron count was compared. Brachial plexus avulsion induced sustained death of motor neuron cells in the vehicle control rats, leaving only around 23.1% surviving neurons at 6 months (Figure 2A–C). The treatment groups showed significantly higher numbers of surviving motor neurons compared to the avulsion control group (*p* < 0.05) (Figure 2B). However, the 7-nitroindazole treatment group’s surviving motor neuron counts were significantly higher than those of the SP600125-treated group significant (57.57 ± 2.88 vs. 40.54 ± 2.03; *p* < 0.05). In addition, the profiles of surviving motor neurons in treatment groups were generally larger than those in the vehicle-treated groups. C7 ipsilateral segments showed the highest loss of motor neurons relative to contralateral sides when compared to C6 and C8 segments across all groups (Figure 2D).

### 3.4. Avulsion Causes Longstanding Severe Ultrastructural Changes in Motor Neurons

In the contralateral side, neurons, ultrastructural features of the organelles such as mitochondria, lysosomes, Nissl bodies (flattened endoplasmic reticulum with bound ribosomes), and lysosomes were evenly distributed, packed, and preserved in the cytoplasm (Figure 3A–C). After surviving for 6 months post-brachial plexus avulsion without treatment, the motor neurons in the vehicle-treated group exhibited morphological changes indicative of neurodegeneration as well as ultrastructural features of cellular death such as mitochondrial morphology disruption, chromatolysis, invasion by astrocytes, as well as disruption of membrane-bound organelles such as the Golgi apparatus, endoplasmic reticulum, and accumulation of phagocytic bodies (Figure 3D–F). Many neurons sampled from this group had fewer organelles as well as appeared necrotic and some were pre-apoptotic with a narrow rim of cytoplasm remaining around the nucleus. These motor neurons had cellular membrane disruptions and were close to astrocytes and microglia suggestive of invasion and phagocytosis of cellular debris (Appendix A). On the other hand, motor neurons in the neuroprotective agent-treated groups had close relations with glial cells, especially astrocytes but did not exhibit phagocytic features (Appendix A). Immunofluorescence images showed that, at 6 months, the ipsilateral ventral horn of vehicle-treated rats still had significantly high levels of astrocytes activation as shown by GFAP optical density (Appendix A).

In the 7-nitroindazole (Figure 3G–I) and SP600125 (Figure 3J–L) treated rats, most surviving motor neurons generally preserved much of their organelles such as the Golgi apparatus, endoplasmic reticulum, and bound ribosomes although they are disorganized indicative of injury and their lumina irregular when compared to the contralateral side. The nuclear envelope, which forms a selective barrier between the nucleus and cytoplasm, can develop intricate morphological changes following some physiological and pathological inducements [38]. In the 7-nitroindazole-treated rats, most of the nuclei possessed highly invaginated nuclear envelopes forming intricate “nucleoplasmic reticula” which formed elaborate networks of tubules and sheets of both inner nuclear membrane (INM) and outer nuclear membrane (ONM). In extreme cases, the nuclear mass appeared as a multilobed mass in close apposition as the invaginations traversed the nucleus completely (Appendix A). A scheme of classifying nuclear nucleoplasmic reticulum (NR) devised by [39] distinguished Type I NR (an invagination composed solely of the INM) from a type II (invagination composed of both the INM and ONM enfolding a diffusion-accessible cytoplasmic core). In the 7-nitroindazole-treated group, the type II NR was predominant with isolated cases of type I coexisting with type in a quarter of the rats’ motor neurons. Nuclear envelope blebbing is one of the hallmarks of apoptosis mediated by the caspases [40]. At 6 months after an avulsion, 7-nitroindazole-treated rats had an intact nuclear double-walled envelope with isolated blebbing of the ONM in a few neurons. In the JNK inhibitor SP600125-treated group, the rats’ motor neuron nuclei showed the mildest form of nuclear envelope undulations with the majority of the nuclear envelope outlines being smooth with less ONM nuclear blebbing than the 7-nitroindazole treatment group.

The mitochondrial morphology is a reliable indicator of the relative health of the host cell. We adopted a health scale of 1 to 5 (Figure 4A,B), whereby 5 represents mitochondria of healthiest possible appearance (distinct and well-packed cristae, electron-dense, completely intact inner and outer membranes) while 1 is degenerate [30]. Most of the mitochondrial matrix in the vehicle group were electron-lucent, degenerate with the mitochondrial cristae severely fragmented, warped, irregular, and the organelle outline generally spherical. The 7-nitroindazole and SP600125-treated rats showed significant preservation of mitochondrial health relative to the vehicle control group (*p* < 0.05). In addition, there was also a characteristic presence of numerous elongated, electron-dense mitochondria with closely packed cristae throughout the cytoplasm of almost all 7-nitroindazole and SP600125-treated rats compared to the control group (Figure 4A). Generally, the 6 months post avulsion mitochondria in the SP600125-treated rats motor neurons were less healthy compared to those of the 7-nitroindazole rats (Figure 4B). The mitochondria in 7-nitroindazole and SP600125-treated rats were significantly larger and numerous compared to the smaller and circular in the vehicle-treated controls.

### 3.5. Downregulation of nNOS by 7-Nitroindazole Preserved Synaptic Inputs on Alpha Motor Neurons

Brachial plexus roots avulsion causes synaptic stripping from motor neurons within the first month [2,41]. The ultrastructural evaluation of the contralateral and control rats’ ventral horn alpha motor neurons (number of synapses per 100 µm of the membrane) revealed no differences in the frequency of synaptic terminals on their membranes (Figure 5A). As identified previously by Conradi [29], the synaptic contacts were classified as F-, S-, or C-type, representing the inhibitory (GABA and/or glycine), excitatory (glutamatergic), and cholinergic inputs, respectively [42]. Our results show that there were more F-type synaptic inputs followed by S- and then fewer C-type across all groups (Figure 5). In the brachial plexus roots avulsed vehicle control group, there was a widespread detachment of the synaptic inputs from the surface of the alpha motor neurons eliminating nearly three-quarters of the contacts and in many cases, astrocytic processes could be seen in the space created by the retractions (Figure 5). On the other hand, 7-nitroindazole treatment significantly preserved the F- and S-type synaptic apposition and numbers within two-thirds of the control covering (Figure 5).

### 3.6. Ultrastructural Observations of Axons in the Vicinity of Motor Neurons

Semi-quantitative scoring revealed a greater extent and more widespread axonal pathology in the vehicle-treated avulsion group compared to the avulsion with neuroprotective groups. Contralateral, non-avulsed sides did not exhibit any axonal degenerative changes. Electron microscopic evaluation showed the presence of predominantly small and medium-sized myelinated fibers, as well as small unmyelinated axons along with Schwann cell nuclei and debris in the vehicle, treated avulsed ventral horns (Figure 6A). There were also extensive areas of the endoneurium laden with collagen fibrils deposits. In the rats that received neuroprotective treatments, the axons near the alpha motor neurons were predominantly larger caliber with orderly sheathing as well as less debris (Figure 6B).

### 3.7. Neurovascular Unit Preserved by Neuroprotective Treatment

In the contralateral side ventral horn, squamous endothelial cells lined microvessels and were covered by a continuous basement membrane. The profile of the endothelium was thin in areas further from the bulging nucleus with a few small granules inside the cytoplasmic rim. The tight junctions between apposing membranes were clear. Less than a third outer circumference of the endothelial cells was covered by pericytes, Figure 7. In the vehicle control group, the endothelial wall was uneven, large vacuoles present, and the tight junctions less electron-dense and some were disrupted. The pericytes were swollen and covered between 30% and 50% of the circumference of the endothelial circumference and were not significantly different from JNK inhibitor SP600125 or 7-nitroindazole-treated rats.

Astrocytes have scant electron-dense material in their cytoplasm along with less dense mitochondrial membranes. Six months after an avulsion, we found many astrocytes near degenerating motor neurons and microvessels. The perivascular astrocytes were still swollen with numerous end-feet compared to observations made in the control contralateral sides. There were no significant differences in the cross-sectional area of astrocyte endfeet between control and treated groups. However, the astrocytic mitochondria in treated groups had better organized cristae than vehicle-treated control.

### 3.8. Functional Recovery at Six Months

#### 3.8.1. Terzis Grooming Test

Results of the Terzis grooming test showed that rats in the neuroprotective treatment groups (7-nitroindazole and SP600125) scored progressively better from the second week up to 6 months compared to those in vehicle control (Figure 8A). Qualitatively, however, the improved movements in the neuroprotective agent-treated groups were largely because of gross motor movements of the proximal forelimb muscles. There was no significant difference between 7-nitroindazole and SP600125-treated rats’ grooming abilities (*p* > 0.05)

#### 3.8.2. Grid Walk Test

Grid walk test results showed subtle deficits in fine motor function not captured in the Terzis grooming test. The results at 6 months showed that the vehicle group (1.3333 ± 0.05), SP600125- (1.6667 ± 0.08) had severe challenges in forepaw placement and balance while significant improvement was noted in the 7-nitroindazole-treated rats (2.33333 ± 0.1) (Figure 8B). Pre-surgery (Pre-OP) rats did not show any significant impairment in runway crossing at both time points. The 7-nitroindazole-treated rats were able to move the forepaws on placement while half of the rats in controls showed no movement of forepaws as they were clasped in a fully flexed position. There was a statistically significant difference between the groups 3 (*p* = 0.02) and 6 months after injury (*p* = 0.02) although there was a general tendency at 6 months for 7-nitroindazole rats to perform better than other treatment regimens (*p* > 0.05).

## 4. Discussion

There are a paucity of morphological data on the long-term effects of brachial plexus roots avulsion and neuroprotective therapies targeted at the ventral spinal cord components. Therefore, ultrastructural and functional changes related to the avulsion and neuroprotective therapies targeted at the JNK c-Jun and nNOS signaling pathways were evaluated at 6 months post-avulsion. This study demonstrated that without treatment the rats experienced avulsion induced significant glial activation, sustained motor neuron loss, synaptic stripping, chromatolysis, substantial ventral horn atrophy as well as spontaneous but limited functional recovery characterized by poor forepaw dexterity. One month of daily neuroprotective treatments targeted at inhibiting the c-Jun JNK (SP600125, 25 mg/kg) and nNOS (7-nitroindazole, 30 mg/kg) signaling pathways significantly reduced c-Jun and nNOS proteins, respectively along with nitric oxide synthesis, oxidative stress, motor neuron apoptosis as well as promoting ipsilateral forelimb function. Long-term effects of these treatments spared some motor neurons, and ventral horn size at 6 months post-avulsion. Ultra-structurally, neuroprotective treatment was characterized by elongated mitochondria and reticularized nuclei in motor neurons whilst the vehicle-treated rats were saddled with apparent degenerative changes disrupting mitochondrial health and increased incidence of astrocytic invasion of motor neurons.

The loss of motor neurons following brachial plexus roots avulsion is long established in adult mammals [6,43]. Our previous studies and those of others established that the loss of motor neurons is mainly robust during the first month with the peak being observed during the second and third weeks post-avulsion leaving around 40% [6,7,44,45]. Also, the first 10–12 days is the most critical period for the survival of rat motor neurons [44]. Later studies have also noted a progressive loss of motor neurons and that the loss slowed down up to three months in rats with nearly a third of the motor neurons remaining relative to the contralateral side [46,47,48,49]. The present study established that the ipsilateral motor neuron loss in avulsed rats’ spinal segments continues, albeit slowly, at 6 months leaving around a fifth of the contralateral side. This trend is consistent with earlier studies in rats [50,51], cats [52], rabbits [51], and monkeys [43]. Also, at this time point, significant atrophy of the ventral horn of the ipsilateral spinal cord segments had set in. Neuroprotective treatments (SP600125 and 7-nitroindazole) given once daily during the first post-avulsion month preserved some motor neurons at 6 months compared to vehicle controls. The neuroprotective treatments targeted the period when most of the motor neurons are known to die via apoptosis. During this month-long period, both SP600125 and 7-nitroindazole significantly down-regulated pro-apoptotic proteins Bim and Caspase 3 as well as its cleaved version. In addition, nitric oxide synthesis and nNOS activity were suppressed as evidenced by attenuated total nitric oxide production as well as NADPH-d activity in SP600125 and 7-nitroindazole-treated rats. These results demonstrate that early but long-term, neuroprotective treatments rescued motor neurons from death and slowed down their death at 6 months. This is in contrast with a previous study in which single-dose neurotrophic factor treatments (brain-derived neurotrophic factor (BDNF) and ciliary neurotrophic factor (CNTF) failed to rescue motor neurons at 6 months after the avulsion lesion [51]. In the latter study, however, it was likely that a single-dose treatment was not sufficient to afford long-term neuroprotection like that observed when neuroprotective therapies are sustained like in the current study. Clinically, patients with brachial plexus injuries suffer multiple injuries that delay treatments such as surgical repair or reimplantation [37]. Most motor neurons would die during this time without neuroprotective treatment thereby leading to a high residual disability burden. Therefore, available neuroprotective treatments should be offered as early as it is safe to do so and the search for neuroprotective treatments must be expedited to prevent permanent loss of function among patients.

In the present study, without treatment, almost 80% of the motor neurons had died at 6 months after avulsion leading to the observed profound loss of fine motor forelimb function. However, in the rats that received a month of neuroprotective treatments, there was also the preservation of ventral horn size (area) and function. The preserved ventral horn mass suggests that other components beyond motor neurons were also saved. As noted previously, neuroprotective treatments given early after injury tend to protect motor neurons that would have died anyway, thereby giving them a chance to survive, sprout and re-innervate their targets [53]. There is a need to buy time in the clinic so that axotomized motor neurons can sprout and reach their targets such as muscles before they become fibrotic and degenerate too [54,55]. The current neuroprotective treatments, 7-nitroindazole and SP600125 given long term, did partially preserve motor neurons, ventral horn size, and hence, function. Also, the neuroprotective-treated groups had better profiles of axons in the vicinity of the motor neurons in the ventral horn suggesting their usefulness to improved functions of the forelimbs at 6 months. However, the recovery in the control rats was mainly observed in proximal muscles and not distal forelimb that would have accounted for fine motor movement. This observation is concordant with earlier reports whereby cervical roots avulsion in rats led to functional recovery only in the proximal limb muscles that are closer to the injury site and hence easier to reinnervate [51,56,57,58]. Also, long-term preservation of grey matter mass observed in this study implies that monitoring of tissue-specific cord pathology can act as a potential biomarker and helps to support more efficient targeting and monitoring of neuroprotective (i.e., gray matter) agents. Ventral horn atrophy is a poor prognostic factor [59] while preservation can support late treatments such as cellular transplantations [14]. Future studies need to continue unraveling the long-term morphological status of targetable components of the ventral horn such as interneurons, glial cells, and the associated vascular system.

Ultrastructurally, vehicle-treated rats’ surviving motor neurons exhibited neurodegenerative changes such as chromatolysis, membrane integrity disruption, mitochondrial cristae disruption, and invasion by astrocytes at 6 months after avulsion. The loss of cell membrane integrity was prominent in these rats 6 months post-avulsion indicating continued loss of motor neurons. The cytoplasmic contents released into surrounding tissue causes chemotaxis and recruitment of pro-inflammatory cells such as microglia and astrocytes observed around motor neurons. Although microglial cells have largely been considered phagocytic, recent studies have demonstrated the phagocytic nature of astrocytes through synapse elimination and clearing apoptotic cells [60,61,62]. In the current study, vehicle-treated rats had evidence of astrocytic invasion, synaptic stripping, and intervening spaces between motor neurons and synapses were occupied by astrocytic processes. Astrocytes replace microglia from the surface of motor neurons and synapse recovery following successful regeneration in the periphery coincides with the disappearance of astrocyte wrapping [63]. In the present study, however, at 6 months, astrocytic processes were still present on the surfaces of surviving motor neurons, especially those with unhealthy ultrastructural appearance. According to previous studies, avulsion causes synaptic stripping coupled with preferential loss of the excitatory S-types buttons [42,52]. Similar observations were made at 6 months after avulsion in this study. However, neuroprotective treatments appeared to have preserved F-, S- and c-type synapses and showed no evidence of dead/dying motor neurons being invaded by astrocytes. In addition, astrocytes observed near motor neurons did not show signs of engulfment or phagocytic activity towards motor neurons. As noted by [64], astrocytes could have promoted long-term synaptic preservation or reinnervation of motor neurons after brachial plexus avulsion [43]. The preservation of synaptic boutons by 7-nitroindazole or SP600125 treatment suggests that nitric oxide suppression offered protection from synaptic withdrawal. C-Jun is a major subunit of Activator protein 1 (AP-1) transcription factor that is upstream of the nNOS gene [65,66] and thus could also have influenced the production of nitric oxide. Also, neuronal nitric oxide is “necessary” to induce withdrawal of synaptic terminals on injured motor neurons by prolonged activation of the soluble guanylyl cyclase (sGC)/protein kinase G (PKG) pathway and RhoA/Rho kinase (ROCK) signaling [67,68,69]. Therefore, its suppression could have been neuroprotective and spared synapses [70]. However, it remains unclear if this is what happened in the present observation, and if so, by what mechanism activated glial cells perform a restorative process, especially after long-term neuroprotective therapy.

The observed dynamic alterations to the nuclear envelope comprising a network of penetrating and branching invaginations known as the nucleoplasmic reticulum were present in some 7-nitroindazole-treated rats’ motor neurons. The existence of the nucleoplasmic reticulum in some cells such as fibroblasts, breast cancer, and endometrial cells is now widely accepted but its precise function is not fully understood [39,71]. Speculations are that it enables the communication between the cytoplasm and nucleoplasm by increasing the interface between these two environments as well as participate in calcium signaling, gene expression, transport, and nuclear lipid metabolism [38]. Since the activation of the nNOS signaling pathway, blocked by long-term 7-nitroindazole treatment, is downstream of calcium ion influx into the motor neuron, it could be possible that the nucleoplasmic reticula observed in the nuclei of rats in this group were a compensatory response to the treatment. The objective nuclear morphology quantification and implications of such a response require further study.

Neurons respond to neuronal stress by activating signaling pathways such as Ca^2+^, CREB, PGC-1α, and NF-κB that stimulate mitochondrial biogenesis and cellular stress resistance [72]. The failure of the antioxidant defenses contributes to motor neuron apoptosis and this was prominent in this study’s vehicle-treated rats. The 7-nitroindazole-treated rats’ motor neurons tended to have elongated and branched mitochondria with closely packed cristae when compared to the SP600125 or control rats. The vehicle-treated rats had the smallest circular mitochondria with disrupted cristae. Generally, most earlier reports suggested that long mitochondria (fusion) protect against stressors until they are unable to restore mitochondrial homeostasis. At this breaking point and beyond, the mitochondrial fission predominates leading to the removal of damaged mitochondria as well as destabilizing the motor neurons and initiates a proapoptotic signaling cascade [73,74]. In addition, nitric oxide production by nNOS activation is known to induce mitochondrial cytochrome c release and down-regulation of Bcl-2 leading to caspase activation [75,76]. In earlier studies, cell death repressor Bcl-2 expression was increased following neuroprotective treatments including 7-nitroindazole [75,76,77]. Therefore, in 7-nitroindazole-treated rats, mitochondria could have assumed cellular stress resistance leading to motor neuron and functional preservation. However, further studies are required to verify the role of Bcl-2 post-avulsion and neuroprotective treatments.

Endoplasmic reticulum morphology and its relationship with other organelles such as the nucleus and mitochondria are intricately related to its function. Previous studies have reported its disruption following avulsion [78]. An extensive survey of ultrastructural endoplasmic reticulum morphology at 6 months after avulsion with or without neuroprotective treatment has not been reported previously. For earlier time points, endoplasmic reticulum swelling beyond 90 nm after axotomy was recorded and increased tethering to mitochondria increases chances of survival and axon regeneration [79,80]. However, this was not observed in this study at 6 months. Electron microscopy studies have shown extensive endoplasmic reticulum pathologies such as chromatolysis, membrane disruption, and displacement in motor neurons [81,82]. Evidence of chromatolysis was prevalent in motor neurons of vehicle-treated rats 6 months after avulsion while the surviving motor neurons of treated rats showed signs of recovery or preservation. This was consistent with earlier findings of Price and Porter [83] who reported that reformed granular endoplasmic reticulum produces materials important for functional recovery. A combination of ER stress and mitochondrial dysfunction involving reactive oxygen species generation and cytochrome c release can result in apoptotic neuronal cell death [84]. All these cascades are blocked by either SP600125 or 7-nitroindazole treatments and this explains why these long-term neuroprotective treatments preserved some ultrastructural morphology of the organelles inside motor neurons.

## 5. Conclusions

The progressive loss of motor neurons and ventral horn atrophy continued to be detected at 6 months after brachial plexus roots avulsion. The motor neurons were degenerative with disrupted cellular membranes, Nissl substance, and mitochondria as well as invasion by astrocytes. Neuroprotective treatments using SP600125 or 7-nitroindazole could provide a basis for early and long-term neuroprotective functions as well as delayed interventions geared to repair brachial plexus roots avulsions.

## Figures and Tables

**Figure 1 cells-10-01614-f001:**
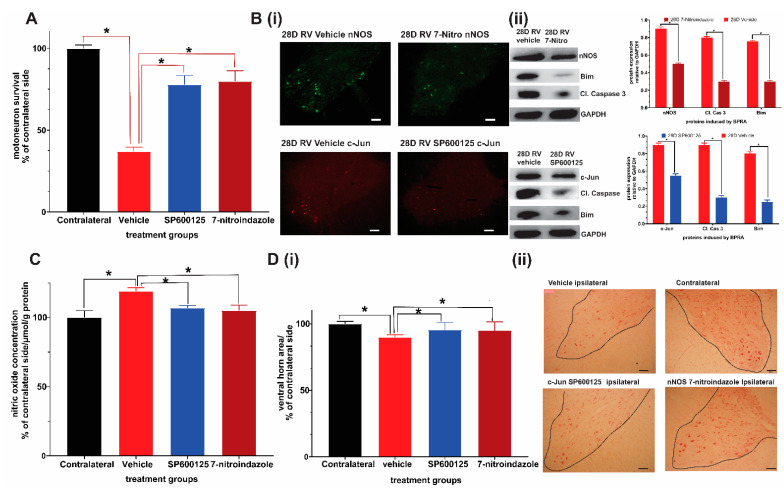
Neuroprotective effects of 28-day daily intraperitoneal injections of c-Jun/JNK inihibitor SP600125 and nNOS inihibitor 7-nitroindazole (7-nitro). (**A**) Comparison of ventral horn motor neuron survival at 28 days (1 month) after avulsion between the control and SP600125 or 7-nitroindazole treatment groups (*p* < 0.05, *n* = 5/group). (**B**) Representative immunofluorescence shows that SP600125 and 7-nitroindazole treatment suppressed the expression of c-Jun and nNOS, respectively at 28 days after avulsion relative to vehicle treatment. Representative Western blots show that SP600125 and 7-nitroindazole treatment suppressed the expression of c-Jun and nNOS, respectively at 28 days after avulsion relative to vehicle treatment. Tissue lysates were obtained from ipsilateral ventral horns (*n* = 5; *p* < 0.01). The neuroprotective treatments also suppressed apoptosis of motor neurons by inhibiting the overexpression of components of the mitochondrial pathway of apoptosis namely Bim, and cleaved caspase 3 compared to that observed in vehicle-treated control rats. (**C**) Treatment with SP600125 and 7-nitroindazole significantly reduced oxidative stress as indirectly measured by the total nitric oxide production compared with the controls. (**D**) The sizes of the corresponding ipsilateral/contralateral areas were computed by planimetry (**i**). There was significant atrophy of ipsilateral ventral horns of vehicle-treated rats relative to contralateral side (**ii**). Data expressed as Mean ± standard deviation (SD), as a percentage of values of the corresponding contralateral side. Comparisons by one-way analysis of variance (ANOVA) whereby * *p* < 0.05, *n* = 5 rats.

**Figure 2 cells-10-01614-f002:**
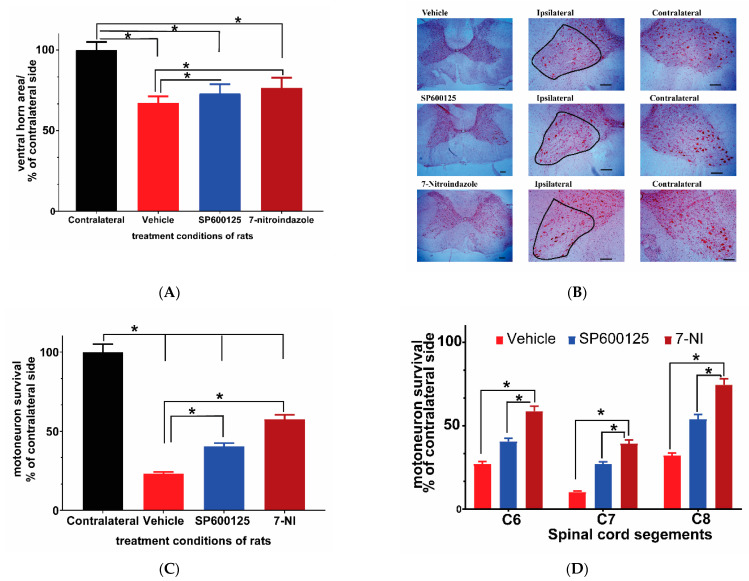
Ventral horn area and motor neuron survival in C6, 7, and 8 cervical segments 6 months after avulsion. (**A**) Comparison of ventral horn size (area) 6 months after avulsion between the control and SP600125 or 7-nitroindazole (7-NI) treatment groups (*n* = 4/group). The sizes of the corresponding ipsilateral/contralateral areas were computed by planimetry. There was significant atrophy of ipsilateral ventral horns of vehicle, 7-nitroindazole, and SP600125-treated rats. Data expressed as mean ± SD, as a percentage of values determined in the corresponding contralateral side. Comparisons by one-way ANOVA followed by Tukey’s honestly significant difference (HSD) whereby * *p* < 0.05, *n* = 4 rats. (**B**) Neutral red-stained right ipsilateral and left contralateral ventral horns of the cervical spinal cord showing motor neurons (black outline) as seen under the light microscope (40× magnification) (scale bar, 100 µm). The representative images of the left and right ventral horns of each of the groups: vehicle, SP600125, and 7-nitroindazole are shown. (**C**) Survival of motor neurons (expressed as the mean percentage of the contralateral side 6 months after avulsion) in control and the neuroprotective agent (7-nitroindazole and SP600125) treated rats. Data reported as mean ± SD, as a percentage of values determined in the corresponding contralateral side. * *p* < 0.05, *n* = 4 rats; Tukey’s HSD * *p* < 0.05). (**D**) Motor neuron survival in C6, 7, and 8 cervical segments 6 months after avulsion. The magnitude of motor neuron loss was greatest in the vehicle-treated rats compared to groups that received neuroprotective treatments. Generally, the 7th cervical segment had the least number of surviving motor neurons across all groups. Data expressed as mean ± SD, as a percentage of values determined in the corresponding contralateral side. * *p* < 0.05, *n* = 4 rats.

**Figure 3 cells-10-01614-f003:**
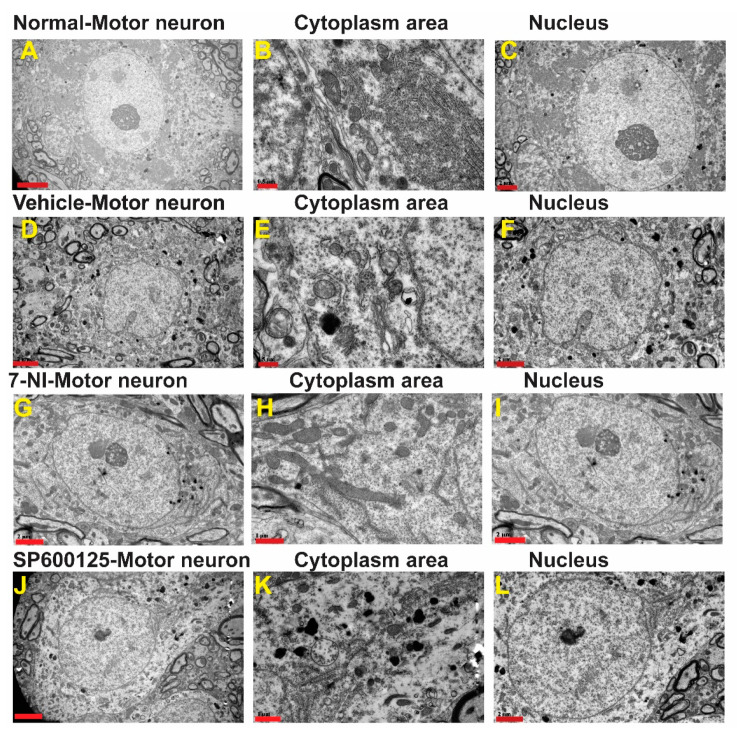
Ultrastructure of representative α-motor neurons. **A**–**C**: **A**—Ultrastructure of representative normal α-motor neuron from the contralateral side showing organelles (**B**) and nucleus (**C**). 1 µm and 2 µm; **D**–**F**: **D**—Ultrastructure of representative α-motor neuron from the ipsilateral side of a vehicle-treated rat showing organelles in cytoplasm (**E**) and nucleus (**F**). Scale bars = 0.5 µm and 2 µm; **G**–**I**: Ultrastructure of repesentative α-motor neuron from theipsilateral side showing 7-nitroindazole showing organelles in cytoplasm (**H**)—Ultrastructure of representative α-motor neuron from the ipsilateral side of a 7-nitroindazole-treated rat showing organelles (**H**) and nucleus (**I**). Scale bar = 1 µm and 2 µm; **J**–**L**:**J**—Ultrastructure of representative α-motor neuron from the ipsilateral side of as SP300125-treated rat showing organelles (**K**) and nucleus (**L**). Scale bar = 1 µm and 2 µm.

**Figure 4 cells-10-01614-f004:**
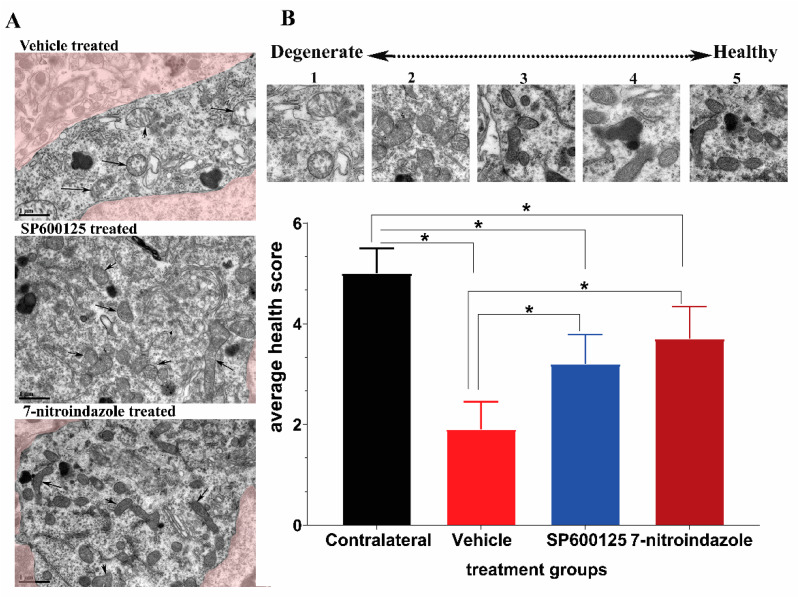
Representative transmission electron microscopy (TEM) images showing mitochondrial morphology. (**A**) shows the mitochondria of motor neurons treated with vehicle, SP600125, and 7-nitroindazole. (**B**) top part shows a health scale of 1 to 5, whereby 5 represents mitochondria of healthiest possible appearance (distinct and well-packed cristae, electron-dense, completely intact inner and outer membranes) while 1 is degenerate. Magnification 13,500×. Scale bar = 1 µm. Differences between contralateral side and vehicle or 7-nitroindazole and SP600125-treated rats were significant at * being *p* < 0.05 using one-way ANOVA followed by Tukey’s HSD.

**Figure 5 cells-10-01614-f005:**
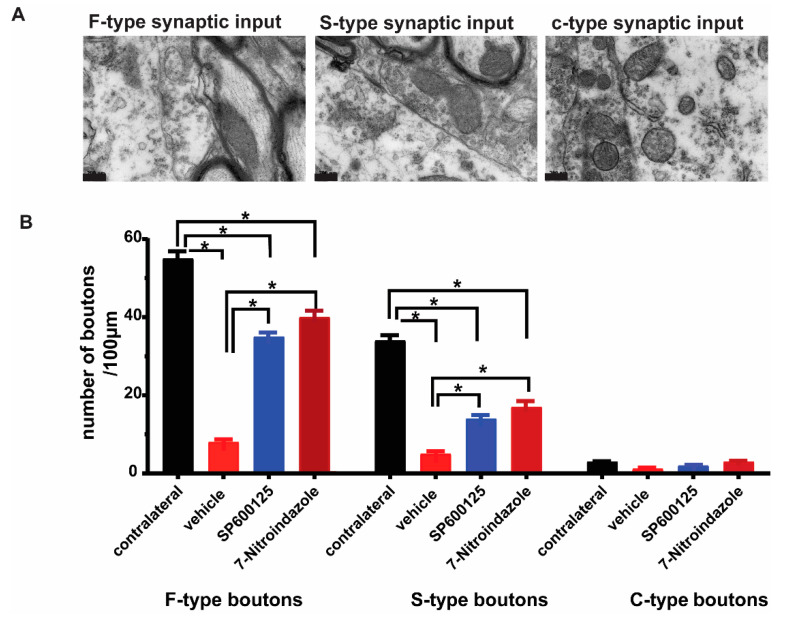
Representative TEM images of the ultrastructure of synapses F-, S- and c-type synapses apposed to α motor neurons. (**A**,**B**). Types and distribution of synaptic buttons types per 100µm on the surface of the vehicle, 7-nitroindazole, and SP600125-treated rats’ motor neurons. Normal refers to the contralateral side without avulsion. Scale bar: 200 nm. (**C**). Ultrastructure of synapses apposed to α motor neurons showing near normal input apposition on the surface of 7-NI (7-nitroindazole) α motor neuron, near normal and partial retraction (SP600125-treated) and retracted terminal intermingled with an astrocyte projection in the vehicle (avulsion only) rats. Motor neuron; (**C**), SP600125-treated rat shows retracted terminal intermingled with an astrocyte projection following ventral root avulsion. (**D**). Distribution of synaptic buttons per 100 µm, as well as synaptic covering on the surface of the vehicle, 7-nitroindazole, and SP600125-treated rats’ motor neurons. Normal refers to the contralateral side without avulsion used as control. The colorized part is the α motor neurons. Values are expressed as the Mean ± SD. Comparisons between groups were by One-way ANOVA whereby * denotes *p ≤* 0.05.

**Figure 6 cells-10-01614-f006:**
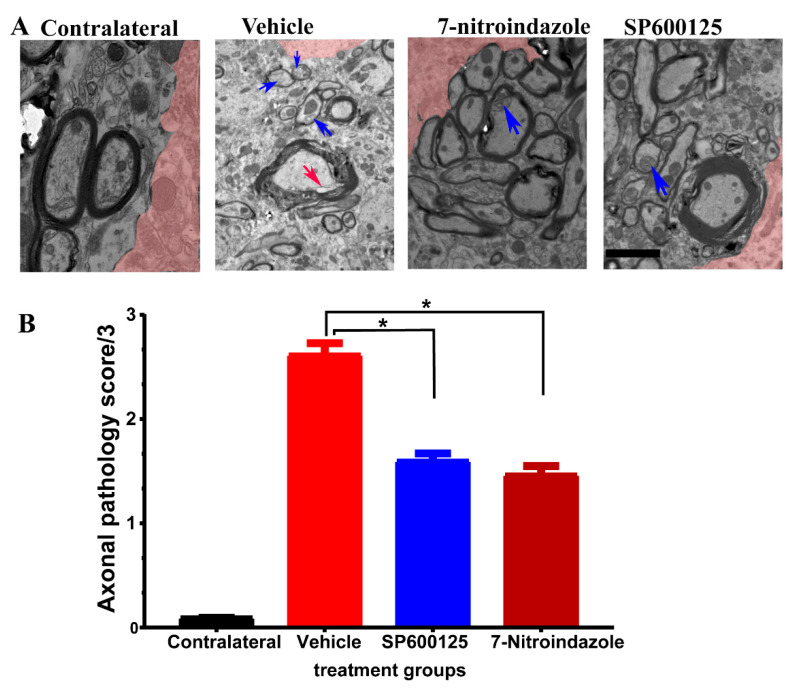
Axonal pathology scores at 6 months after avulsionfollowed by 1 month vehcle and neuroprotective treatment. (**A**) Representative TEM images showing the ultrastructure of axons in the vicinity of α motor neurons. The colorized part is the α motor neurons. (**B**) Axonal pathology scores. Values are expressed as the mean ± SD. Comparisons between vehicle and treatment groups were by One-way ANOVA whereby * denotes *p* < 0.05. The red arrow shows vacuole while blue arrows depict disintegration and edema.

**Figure 7 cells-10-01614-f007:**
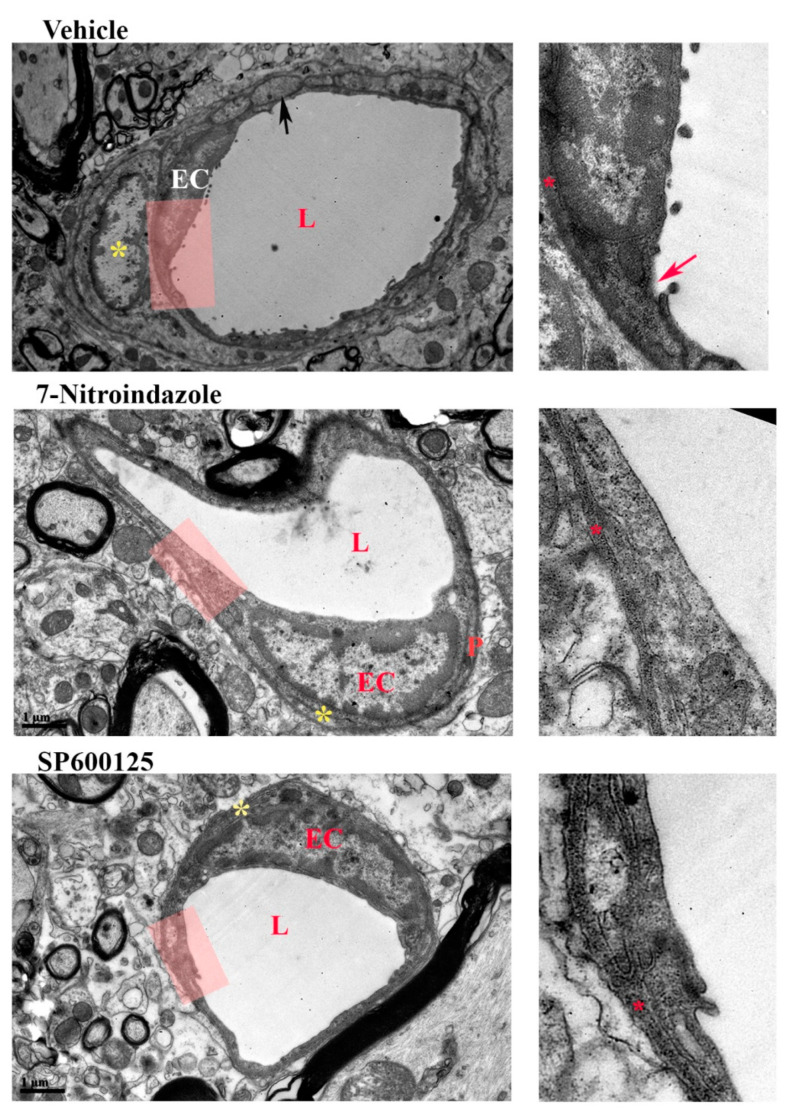
Neurovascular unit. Electron microscopy showing representative spinal cord microvessels in the left C7 ventral horn of rats injected with either vehicle, JNK inhibitor SP600125, or 7-nitroindazole. Magnification 9700× and 18,500×. Scale bar: 1 µm. L = lumen, EC = endothelial cell. Red arrow- disruption of endothelial membrane. Red star (*) = basement membrane. Yellow star * = pericyte.

**Figure 8 cells-10-01614-f008:**
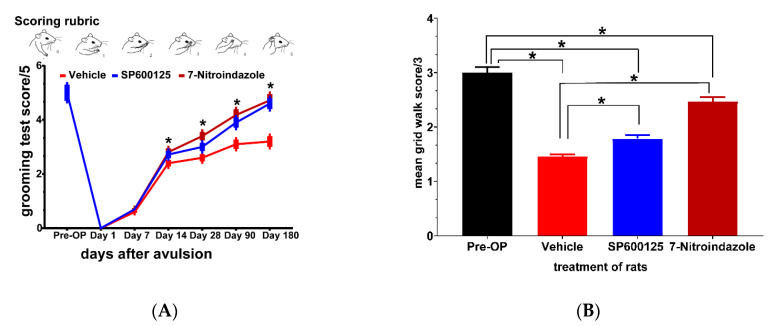
Neuroprotective treatments enhance recovery of forelimb function after ventral spinal root avulsion. (**A**) Positions of the forelimbs and corresponding scores (0–5) of the Terzis grooming test; (**B**) grid walk test scores for evaluating impairment in runway crossing. Data are expressed as the mean ± SD (*n* = 4 in each group). For the grooming and grid walk tests * *p* < 0.05, Kruskal Wallis test followed by the Tukey post hoc test was used to compare multiple groups (vehicle versus SP600125 or 7-nitroindazole and SP600125 versus 7-nitroindazole).

## Data Availability

The data presented in this study are available on request from the corresponding author.

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
