# Peer review of "Long-Term Suppression of c-Jun and nNOS Preserves Ultrastructural Features of Lower Motor Neurons and Forelimb Function after Brachial Plexus Roots Avulsion"

_cells, 2021, doi:10.3390/cells10071614_

Round 1
Reviewer 1 Report
This is a very interesting study where the authors have performed a brachial plexus avulsion on rats and assessed the short-term (28 days) and longer-term (6 months) effect of the c-Jun JNK inhibitor, SP600125 and the nNOS inhibitor 7-nitroindazole on motor neuron numbers, oxidative stress markers, morphology of the organelles contained within the motor neurons and the rat’s forelimb function. I recommend this study for publication once the comments below have been addressed. 1. Section 2.5 – Morphometrical and Statistical Analysis. There is no information regarding any statistics in this section. 2. General comments for all figures a. It is often very difficult to read the text, especially on the graphs - can a larger font be used? b. The scale bars and associated text are very small and difficult to read – please consider larger bars, larger text size and choice of colour. c. It was difficult to be certain exactly which groups were significantly different to each other when looking at the graphs. Can the p values for the AVOVA as well as the p values for the post hoc tests be reported so that the readers can be certain of which groups are significantly different from each other? 3. Figure 1B – Quantification of the western blot should be included in addition to the images. 4. Figure 2 a. First two sentences of the legend are repeated. b. Legend refers to arrows but can’t see any on the images. c. The red text colour for the scale bar is very difficult to see. 5. Line 422 - do you mean to refer to K-M not just L-M? 6. Line 433 – do you mean to refer to Supp Fig 2? 7. General query about the statistics used for the behavioural studies – is a one way ANOVA the most appropriate statistic to use here? As rating scales/scoring systems were used to measure the rat’s motor function, isn’t a Kruskal Wallis test more appropriate? The rating scales/scoring systems are ordered categories therefore giving you ordinal data which may be analysed with a Kruskal Wallis test while the one way ANOVA is mainly used when the data is interval and normally distributed.Author Response
Reviewer 1
We greatly appreciate the feedback of this reviewer and take note of the insightful comments.
Specifically,
- The section 2.5 had ben revised to remove the word statistics so that a description of the statistical tools used are described in section 2.11 Statistical analysis section.
- a, b) The images have been revised so that the scale bars and associated text which was too small to read has been revised and where relevant, color changes instituted for better visibility.
- c) We have also revised the significant statistics marks and made sure there is specification of respective P-Values for ANOVA and post-hoc tests as suggested.
- In figure 1B, we have included the WB quantification and associated statistical analysis.
4 a) The repetition in the figure 2 legend has been removed.
- b) Fig 2 has been revised to remove arrows and marked the ventral horn areas bearing motor neurons that are being lost post avulsion in the figure as they have been mentioned in the figure legend.
- c) The red color of the scale bar has been enhanced for easier visibility as suggested.
- The A-C, D-F, G-I and J-L labels of the images have been revised to reflect the correct order and the text corrected as appropriate.
- The reference in line 433 is to Supplementary figure 2 and has duly been amended and corrected in text.
- We agree with the suggestions of the reviewer and have revised the statistical analysis to Kruskal Wallis for ordinal data. The significant changes remain as before.
Reviewer 2 Report
The manuscript titled"Long-term suppression of c-Jun and nNOS preserves ultrastructural features of lower motor neurons and forelimb function after brachial plexus roots avulsion" by Zilundu et al. is a morphological study which investigates the long-term suppression of c-Jun- and neuronal nitric oxide synthase (nNOS) (neuroprotective treatments for one month) on the ultrastructural features of lower motor neurons and forelimb function at six months after brachial plexus roots avulsion. The study is enriched by ultrastructural analisis which represents a value added to the investigation.
The manuscript is accepted in the present form.
Author Response
Reviewer 2
We greatly appreciate the time the reviewer took to review our manuscript. We also appreciate the feedback.
Reviewer 3 Report
The paper by Zilundu et al. aims to determine whether SP600125 or 7-nitroindazole have long term neurotropective effects after brachial plexus roots avulsion (effects measured by observing the ultrastructure of the neuons, motor activity, and investigated at the molecular level by Western Blot analysis of different pro-apoptotic markers).
This is an interesting paper but quantitative data are missing, making difficult to be objective on data analysis:
- example: the authors conclude that the invasion by astrocytes in the avulsion group is increased. However no quantitative data are shown for this parameter in the untreated and treated group, to determine the effect of the drugs on this parameter (eg immunostaining, and/or GFAP western blot analysis). Other examples are given below.
- Regarding the anti-apoptotic effect, the authors should investigate this is more depth as suggested below.
- What are the post-hoc tests used in this study, need to explain in more details what the “*” means (differences between which groups).
Several points that need to be addressed:
Figure 1:
- Figure 1 B: the western blots shown both indicate the expression level of c-Jun Caspase 3 Bim and GAPDH in SP600125 treated group: There is probably a mistake in labelling the Western blot here. The molecular wight of each protein is missing, western blot of several sample would be better. Quantification of the western blot is missing. To make the point for mitochondrial apoptosis more solid other elements of the mitochondrial apoptosis pathway have be investigated such as caspase 9, Bax and Bcl2. Necrosis could also be investigated, and this could be complementary to figure 3 results.
- Figure 1C: statistic showing the difference between vehicle and both treatments should be shown (here only one star is present on the graph, unless one group is significantly different, in that case the text for this figure has to be moderated: line 318-320)
- Figure 1D: the authors should show representative picture of ipsilateral/contralateral ventral horn.
Figure 2: Figure 2A: statistical representation needs to be clarify (* is a P<0.5 and difference between which groups? What kind of post-hoc test has been used?)
Figure3: the authors could find a way to quantify intact and abnormal nuclear envelops as described in the text (as well as quantification for other organelles described in the text), and be able to show actual objective values.
Figure 4B: are the values for SP600125, and 7-nitroindazole significantly different from the vehicle values?
Line 410-411: where is supp Fig3A?
Supplemental Figure 2 not mentioned in the main text.
Figure 5 A: legend on x-axes too small, difficult to read.
Figure 8: try to keep the same colour code between A and B (SP and baseline colours are inverted)
Author Response
Reviewer three
We are very grateful for the insightful feedback from the reviewer. We hereby give a point by point response to the matters that arose during peer review.
Generally, we agree about the suggestions to improve the quality of our images to enable ease of communicating the findings. We have also added an astrocytes GFAP labeling supplementary figure accompanying the electromyograph picture showing astrocytic activation and close proximity of motor neurons at 6 months post-avulsion.
Specifically:
Figure 1
Fig1 B: The representative IF and WB images in Fig 1B have been re-labelled to show nNOS expression in 7-nitroindazole treated rats and c-Jun expression in SP600125 treated rats. It was indeed a case of mislabeling. The quantification of the WB has been included along with relevant analyses. While we could not include other elements of the mitochondrial apoptosis pathway such as caspase 9, Bax and Bcl2, we agree with the reviewer that these are important. As a result, we include this in our discussion as a study limitation and recommendation for future research. Also, we include a discussion of our previous study in which Bcl2 was identified in post avulsion motor neurons. In the same vein, necrosis has also been discussed as mentioned above.
Fig 1C. The statistic showing the significant difference between the vehicle and both treatments has been revised to reflect that in the figure legend as well as in the text lines 318-320 as suggested. We hope that the new revision conveys this meaning.
Fig 1D: The representative pictures of ventral horn sizes have been included in Figure 1D and has been divided as D (i) and (ii).
Figure 2: In figure 2A, the area comparisons were done using ANOVA and a Tukey's honestly significant difference (HSD) post hoc test was used showing significant differences between the corresponding contralateral side and ipsilateral sides of the vehicle, 7-NI and Sp600125 treated rats at 6 months. The graph has been revised to show that ALL GROUPS suffered significant atrophy and that vehicle was significantly worse than that of the two groups that received neuroprotective treatments and that the difference between 7-NI and SP600125 treated rats was not significant. The text has also been re-worded to reflect this.
Figure 3: We had included the nuclear morphology as a supplementary file. The ER size data did not exceed 90nm abnormality mark and is excluded as discussed. Text has been adapted to include this data. We have also discussed the need for objective data as a limitation and recommendation for future studies.
Figure 4: The values of neuroprotective treatments were indeed significantly different from the vehicle ones and the figure marks have been revised to show this as described in the text.
Line 410-411, the supplementary figure had been added as a separate pdf file. The image has also been added now for your perusal with adaptations as has been suggested for other figures in general. Supplementary figure 2 has duly been mentioned in text.
Figure 5 The x-axis have been revised and font increased to be easily read.
Figure 8, The coloring of bars has been made uniform throughout the figures.
Overall, we greatly appreciate the thorough feedback and the insights gained from this review. We sincerely hope that our manuscript has now improved for consideration for publication in this esteemed journal.
Round 2
Reviewer 3 Report
The authors answered to all my concerns, and the paper can be published as it is.